# OpenReview forum: "JailBound: Jailbreaking Internal Safety Boundaries of Vision-Language Models"
_NeurIPS.cc/2025/Conference — NeurIPS 2025 poster_

### Official Review · Reviewer_P9we · 2025-06-29

**Clarity:** 2
**Significance:** 2
**Originality:** 3
**Rating:** 4
**Confidence:** 3

**Summary:**

This paper introduces JailBound, a novel jailbreaking framework targeting Vision-Language Models (VLMs) by exploiting internal safety decision boundaries within their latent representations. The approach consists of two main stages: (1) Safety Boundary Probing, which uses logistic regression classifiers to approximate the decision boundary in fusion layer latent space, and (2) Safety Boundary Crossing, which jointly optimizes adversarial perturbations across both image and text inputs to steer the model's internal state toward policy-violating outputs. The method demonstrates better performance compared to existing approaches across six diverse VLMs.

**Questions:**

- Limited discussion is provided on potential defenses or detection mechanisms that could mitigate the proposed attack.
- A thorough computational analysis would help practitioners understand the practical feasibility of the attack and could inform the development of efficient defense mechanisms.

**Ethical Concerns:**

["NO or VERY MINOR ethics concerns only"]

**Final Justification:**

The authors' detailed response and supplementary experiments have substantially strengthened the manuscript and effectively addressed my primary concerns.

**Limitations:**

yes

**Quality:**

3

**Strengths And Weaknesses:**

**Strengths**
- The work addresses a critical safety concern in VLMs, which are increasingly deployed in real-world applications. The high success rates highlight the practical importance of this vulnerability.
- The application of ELK principles to multimodal jailbreaking is novel. The idea of explicitly probing and exploiting safety decision boundaries in fusion layers is original.
- The experimental evaluation is comprehensive, covering six different VLMs across both white-box and black-box scenarios.

**Weaknesses**
- The assumption that safety decisions can be linearly separated in the fusion space may not hold for all VLMs or safety categories. The paper would benefit from discussing when this assumption might fail.
- The experiments are primarily conducted on the MM-SafetyBench dataset, which may not fully represent the diversity of real-world safety scenarios.

---

> ### Author Rebuttal · Authors · 2025-07-31
>
> We sincerely thank you for your constructive questions. Guided by your suggestions, we conduct several new experiments that significantly strengthen our paper's claims and clarity. We appreciate the opportunity to address your points below.
>
> ### **(Weakness 1) On the Limitations of the Linear Boundary Assumption**
>
> **Our Response:** This is a truly brilliant observation. You are right that assuming perfect linear separability is a strong claim, and a nuanced discussion is crucial for scientific rigor.
> Our framework's strength lies in a more general principle: the existence of largely separable regions for safe versus unsafe concepts in the latent space. The linear classifier serves as an efficient and effective surrogate to find a separating hyperplane between these regions. Our method's success does not strictly depend on perfect linearity. As long as the concepts are broadly separable, our method can identify an effective gradient direction. Critically, even if the true decision boundary is curved or non-linear, our optimization (the Crossing stage) operates on the model's actual loss landscape. The probed boundary provides an initial direction, but the final adversarial example is refined based on the model's real, often complex, internal geometry. We have added a discussion to clarify that the linear boundary is a powerful and practical approximation that provides guidance, rather than a rigid prerequisite for our method to function.
>
>
> ### **(Weakness 2) On the Limited Diversity of the Evaluation Dataset**
>
> **Our Response:** We agree that demonstrating the generalizability of our method on a more diverse benchmark is essential. To address this, we have conducted a new set of experiments on the widely-used **JailbreakV-28k benchmark [1]**
>
> **New Experiment: Evaluating JailBound on the JailbreakV-28k Benchmark**
>
> To further validate the generality and robustness of our approach, we evaluated JailBound on the JailbreakV‑28k benchmark using three representative open‑source models (LLaMA‑3.2, Qwen2.5‑VL, and MiniGPT‑4). We compared our results against the SD+Typo baseline, as shown in the table below. JailBound consistently outperforms the baseline across all models, achieving absolute Attack Success Rate (ASR) improvements of +66.4% on LLaMA‑3.2 (89.5% vs. 23.1%), +65.9% on Qwen2.5‑VL (84.2% vs. 18.3%), and +59.2% on MiniGPT‑4 (92.7% vs. 33.5%). These improvements demonstrate that JailBound’s safety boundary–guided optimization is not model-specific but instead generalizes well across different architectures and tasks. This highlights JailBound’s capability to exploit fundamental safety boundary vulnerabilities, enabling state‑of‑the‑art performance on a challenging and diverse benchmark. We have included the full experimental setup and analysis in the Appendix.
>
> | MLLM | Baseline ASR (%)   (SD+Typo) | **JailBound ASR (%)** | **Improvement** |
> | :--- | :---: | :---: | :---: |
> | LLaMA 3.2 | 23.1 | **89.5** | **+66.4** |
> | Qwen2.5-VL | 18.3 | **84.2** | **+65.9** |
> | MiniGPT-4 | 33.5 | **92.7** | **+59.2** |
>
> [1] Luo et al. "JailBreakV: A Benchmark for Assessing the Robustness of MultiModal Large Language Models against Jailbreak Attacks."
>
> ### **(Q1) On potential defenses or detection mechanisms**
>
> **Our Response:** This is an important question. We have added a new discussion on potential defenses and mitigations to our Conclusion. The core of our attack exploits a fundamental geometric property of the model's internal representations. Therefore, effective defenses must operate at this same level. We hypothesize two primary defense strategies:
>
> **Representation-Level Sanitization:** Techniques like representation steering or concept erasing could be adapted to sanitize the fusion layer. These methods would aim to either project harmful representations back into the safe region or to remove the dimensions corresponding to the unsafe concept entirely. This would effectively blur or erase the very decision boundary that JailBound seeks to cross.
>
> **Boundary-Aware Adversarial Training:** A more direct defense would be a form of adversarial training specifically designed to resist boundary-crossing attacks. This would involve generating attacks using a method like JailBound during the training process and explicitly training the model to be robust against perturbations along the normal vector of its own safety boundary. This would harden the boundary, making it more difficult to cross with small, semantically coherent perturbations.
> We believe that investigating these representation-level defenses is a critical next step for the community, and our work provides a clear framework for evaluating their efficacy. This discussion has been integrated into the revised manuscript.
>
>
> ### **(Q2) On Computational Burden and Cost Comparison**
>
> **Our Response:** Thank you for this excellent point regarding the practical computational cost of our method. While JailBound is iterative, its core innovation lies in using a classifier-guided search to achieve efficiency gains over unguided optimization approaches. To provide a transparent comparison as you suggested, we have benchmarked the computational costs.
>
> **New Experiment: Computational Cost Comparison**
>
> We performed a computational cost comparison across different attack methods on LLaMA-3.2-11B, focusing on two efficiency metrics: (1) the average number of iterations required to achieve a successful jailbreak and (2) the average wall‑clock time per sample measured on eight NVIDIA A800 GPUs. The results, summarized in the table below, highlight the substantial efficiency gains of our guided approach.  Compared to the VAJM optimization baseline, which requires roughly 5,000 iterations and 896.67 seconds per sample, and UMK, which requires 2,000 iterations and 577.32 seconds per sample, our proposed JailBound attack succeeds in only about 100 iterations and 68.33 seconds per sample. This represents a 20–50× reduction in iterations and a great improvement in overall attack time. These findings confirm our hypothesis that the probing stage provides effective guidance for crossing the safety boundary, enabling JailBound to achieve both higher success rates and greater computational efficiency. We have added this new table and a detailed discussion on efficiency to Section 4.3 of our revised manuscript.
>
>
> | Attack Method | Avg. Iterations to Success | Avg. Time per Sample (s) |
> | :--- | :---: | :---: |
> | VAJM (Optimization Baseline) | ~5000 | ~896.67s |
> | UMK | ~2000 | ~577.32s |
> | **JailBound (Ours)** | **~100** | **~68.33s** |

---

> > ### Comment · Reviewer_P9we · 2025-08-02
> >
> > Thank you for your effort to provide the detailed response along with the supplementary experiments. These additions have strengthened the paper and addressed my main reservations.

---

> > > ### Author Response · Authors · 2025-08-03
> > >
> > > We appreciate your recognition of our revisions. Your suggestions have greatly contributed to improving our work, we have learned a lot and developed a deeper understanding during the revision process. We are pleased that these changes address your main concerns and hope they also lead to a more positive overall impression of the paper. Thank you again for your valuable time and feedback, and we wish you all the best in both your research and life.

---

### Official Review · Reviewer_9uX9 · 2025-06-30

**Clarity:** 4
**Significance:** 3
**Originality:** 3
**Rating:** 4
**Confidence:** 4

**Summary:**

This paper proposes a jailbreaking method based on the hypothesis that VLMs encode safety-relevant information in their fusion-layer representations. The proposed method works by probing the safety boundary in a VLM's fusion layer and then creating adversarial perturbations to both textual and image inputs, to move the joint representation towards the 'safe' region. Experiments show that the proposed method effectively succeeds in jailbreaking, surpassing other methods in attack success rates.

**Questions:**

Please see the questions above.

**Ethical Concerns:**

["NO or VERY MINOR ethics concerns only"]

**Final Justification:**

The provided additional experimental results addressed my concerns regarding the computational burden and support that linear probing serves as a good approximator of the safety boundary. Therefore, I will increase the score by 1.

Overall, I think this work is a solid technical contribution, however, I think the core methodology is straightforward given previously known approaches (as the other reviewer pointed out), and the insights from the analysis of the safety boundary are somewhat limited (I believe there is room for deeper analyses, e.g., how the boundary emerges in the training dynamics perspective), which justifies staying at the borderline accept and not increasing further.

**Limitations:**

Yes

**Paper Formatting Concerns:**

There are no major paper formatting concerns.

**Quality:**

3

**Strengths And Weaknesses:**

## Strengths

S1: The key idea is clear, straightforward, and sound.

S2: This paper is well-written, and I commend the authors for the clear delivery of core concepts.

S3: Experiments show a clear and significant improvement over other baselines in terms of attack success rates.

## Weaknesses

W1: Computational burden - The proposed method requires >100 steps of gradient-based optimization for each input modality, which may limit the practicality of the proposed method, particularly for larger models.

W2: Regarding W1, I think the comparison of computational costs for the proposed method and other baselines should be shown for a fair and transparent comparison.

W3: The proposed method relies on a key hypothesis that there exists a linear safety decision boundary in VLMs. While the success of the linear probing and the proposed method indirectly supports that this can be true, the hypothesis is not investigated in depth, limiting our understanding of how VLM works. For example, discussing why and how the linear boundary exists (e.g., there could be a connection to the linear representation hypothesis) and why it appears in the fusion layer rather than other components would provide us with deeper insights.

---

> ### Author Rebuttal · Authors · 2025-07-31
>
> We sincerely thank you for your constructive questions. Guided by your suggestions, we conduct several new experiments that significantly strengthen our paper's claims and clarity. We appreciate the opportunity to address your points below.
>
> ### **(Weakness 1 & 2) On Computational Burden and Cost Comparison**
>
> **Our Response:** Thank you for this excellent point regarding the practical computational cost of our method. While JailBound is iterative, its core innovation lies in using a classifier-guided search to achieve efficiency gains over unguided optimization approaches. To provide a transparent comparison as you suggested, we have benchmarked the computational costs.
>
> **New Experiment 1: Computational Cost Comparison**
>
> We performed a computational cost comparison across different attack methods on LLaMA-3.2-11B, focusing on two efficiency metrics:
> (1) The average number of iterations required to achieve a successful jailbreak
> (2) The average wall‑clock time per sample measured on eight NVIDIA A800 GPUs.
> The results, summarized in the table below, highlight the substantial efficiency gains of our guided approach.  Compared to the VAJM optimization baseline, which requires roughly 5,000 iterations and 896.67 seconds per sample, and UMK, which requires 2,000 iterations and 577.32 seconds per sample, our proposed JailBound attack succeeds in only about 100 iterations and 68.33 seconds per sample. This represents a 20–50× reduction in iterations and a great improvement in overall attack time. These findings confirm our hypothesis that the probing stage provides effective guidance for crossing the safety boundary, enabling JailBound to achieve both higher success rates and significantly greater computational efficiency. We have added this new table and a detailed discussion on efficiency to Section 4.3 of our revised manuscript.
> | Attack Method | Avg. Iterations to Success | Avg. Time per Sample (s) |
> | :--- | :---: | :---: |
> | VAJM (Optimization Baseline) | ~5000 | ~896.67s |
> | UMK | ~2000 | ~577.32s |
> | **JailBound (Ours)** | **~100** | **~68.33s** |
>
>
> ### **(Weakness 3) On the Linearity of the Safety Boundary**
>
> **Our Response:** We thank you for this profound question regarding the underlying hypothesis of our work. You are right that a deeper discussion on the existence and nature of the safety boundary is crucial.
>
> First, we wish to clarify a key aspect of our framework: JailBound does not strictly require the safety boundary to be perfectly linear. Our linear probing classifier is used as a highly effective *surrogate* to find an efficient direction for optimization. The core optimization process itself, which perturbs inputs based on the multi-objective loss, remains effective even if the true decision boundary is curved or locally non-linear. This design ensures our method's applicability even as models and their internal safety geometries become more complex.
>
> That said, your question of *why* a linear approximation works so well is fascinating. We believe this aligns with established findings in representation learning. Recent work has significantly advanced the Linear Representation Hypothesis, providing a theoretical basis for our empirical success. Park et al. [1] formally showed that natural categorical concepts correspond to geometric structures like simplices in the representation space, with hierarchical relationships manifesting as orthogonal directions. This aligns perfectly with our observation: "harmful" vs. "benign" can be viewed as high-level semantic categories. It is therefore natural that a separating hyperplane—a linear boundary—would emerge in the model's representation space to distinguish them.
>
> We hypothesize that this feature is particularly strong in the fusion layer because this is where the model integrates and aligns features from both modalities into a unified semantic space, making it the natural locus for these high-level conceptual boundaries to form.
>
> We agree that this is a rich area for future research and have added this discussion, along with the new references, to our paper to provide deeper insight into the principles underlying JailBound.
>
>
>
> To validate that our probing classifier has learned a general representation of "unsafety" rather than just the specific features of our own attack, we tested its ability to identify successful jailbreaks generated by other methods.
>
> **New Experiment 2: Validating the Probing Classifier as a General Surrogate for Jailbreaking**
>
>  We first use three distinct baseline attacks (FigStep, VAJM, and UMK) to generate a set of successful jailbreak samples using the test data on Qwen2.5-VL. Then we extract the internal representations of these successful samples from the model's final fusion layer. Finally, we feed these extracted representations into our probing classifier, which was pre-trained on the same final fusion layer of Qwen2.5-VL using our training data, to evaluate its accuracy in identifying these baseline-generated samples as "unsafe."
>
>
> | Baseline Attack Method (on Qwen2.5-VL) | Prediction Accuracy of Our Qwen2.5-VL Prober |
> | :--- | :---: |
> | FigStep | **96.17%** |
> | VAJM | **97.36%** |
> | UMK | **92.28%** |
>
> As shown in the table. Our probing classifier, trained only to distinguish safe from unsafe inputs in general, achieved high prediction accuracy on these baseline-generated jailbreaks (96.17% for FigStep, 97.36% for VAJM, and 92.28% for UMK). These results indicate that successful attacks produced by different mechanisms converge on a shared, detectable safety boundary within the model, confirming that our framework exploits a general vulnerability. This new experiment and its analysis have been added to the Appendix.
>
> [1]  Park et al. "The Geometry of Categorical and Hierarchical Concepts in Large Language Models."
>
> ### **Other Improvements**
>
> **New Experiment 3: Comparison with Text-only Jailbreak Methods**
>
> To evaluate the contribution of our multimodal approach, we compared JailBound with two representative text‑only jailbreak methods, GCG [28] and AutoDAN [17], on the Llama‑3.2‑11B model.   For these text‑only baselines, we fed their generated adversarial prompts to the VLM with a null (empty) image input, ensuring that only textual perturbations were used. All methods were evaluated on the MM‑SafetyBench test set using Attack Success Rate (ASR) as the evaluation metric. As shown in the table below. GCG and AutoDAN achieved ASRs of 78.27% and 83.52%, respectively, while our full multimodal JailBound attack attained an ASR of 91.40%. These results demonstrate that by jointly optimizing perturbations across both image and text, JailBound achieves a higher success rate than even these text‑only methods. This highlights the critical, synergistic role the visual channel plays in our attack framework, creating vulnerabilities that unimodal attacks cannot fully exploit. This new comparison has been incorporated into our revised manuscript.
>
> | Attack Method | Modality | ASR (%) on MM-SafetyBench |
> | :--- | :--- | :---: |
> | GCG | Text-only | 78.27% |
> | AutoDAN | Text-only | 83.52% |
> | **JailBound (Ours)** | **Image+Text** | **91.40%** |

---

> > ### Comment · Reviewer_9uX9 · 2025-08-02
> >
> > Thank you for your effort in responding to the review with concrete experiments. The provided additional experimental results addressed my concerns regarding the computational burden and support that linear probing serves as a good approximator of the safety boundary. Therefore, I will increase the score by 1.
> >
> > Overall, I think this work is a solid technical contribution, however, I think the core methodology is straightforward given previously known approaches (as the other reviewer pointed out), and the insights from the analysis of the safety boundary are somewhat limited (I believe there is room for deeper analyses, e.g., how the boundary emerges in the training dynamics perspective), which justifies staying at the borderline accept and not increasing further.

---

> > > ### Author Response · Authors · 2025-08-02
> > > **Thank You for Your Feedback!**
> > >
> > > Thank you so much for re-evaluating our work and for the score increase! We are very pleased that our additional experiments successfully addressed your concerns regarding the computational burden and the validity of linear probing.
> > > We also appreciate the reviewer's insightful perspective on the scope of our contribution and agree that a deeper analysis it has already given us a clear direction for our follow-up studies.
> > > We sincerely wish you the very best in your own research endeavors, and a happy, free, and fulfilling life.

---

### Official Review · Reviewer_coGX · 2025-07-01

**Clarity:** 4
**Significance:** 3
**Originality:** 3
**Rating:** 4
**Confidence:** 4

**Summary:**

In this paper, the authors proposed a new jailbreaking attack against VLMs called JailBound. JailBound trains binary probing classifiers to linearly separate safe and unsafe representations on every layer nearly perfectly. The authors believed that these classifiers represent VLMs' latent knowledge about safety regardless of the VLMs' responses to these representations, therefore attacking the classifiers by breaching their decision boundaries is considered a good surrogate task for devising a transferrable jailbreaking. In practice, for each harmful text-image pair, JailBound optimizes for minimal textual and visual perturbations that induce sufficient representation movement in the direction perpendicular to the decision boundaries by combining the losses $L_{sem},L_{align}$ and $L_{geo}$ and using gradient descent and gradient-based token replacement for updates on visual and textual perturbations respectively. The transferability of JailBound was tested by competing with baseline attacks like UMK, FigStep and VAJM on attacking GPT, Gemini and Claude models using the MM-SafetyBench dataset. According to the reported ASRs (measured as $1-$ lexical refusal rates), JailBound transfers significantly better than all baselines consistently with very rare exceptions. The authors didn't put as many efforts in comparing JailBound's whitebox performances with these baselines. They shared an experiment on MiniGPT-4 where JailBound outperformed all the baseline, and compared JailBound with unimodal variants of JailBound in the whitebox setting to confirm that the joint optimization of both modalities is conducive.

**Questions:**

Q1: What comprises $\mathcal{L}_\text{suffix}$ and how was it integrated into the optimization pipeline? The computation of $\mathcal{L}_\text{suffix}$ and $\mathcal{L}_\text{sem}$ where it belong is missing from Algorithm 2 either.

Q2: Is it true that if a jailbreaking attempt is successful, it also successfully confuses the probing classifiers? For the attack baselines, can you apply the classifiers on their text-image pairs? If a significant number of successful jailbreaking attempts don't break the classifiers, then JailBound points out a new attack surface but isn't a framework that covers existing attacks yet.

Q3: For unimodal attacks, in fact there are a lot of text-only jailbreaking methods that are impressively effective. Many of them are tested against VLMs like GPT-4o except that the visual input is empty. If the authors really want to compare with unimodal attacks, I would like to know how well JailBound compares to them.

Q4: Before people apply probing classifiers for jailbreaking, in fact they have been used for defending against jailbreaking. For instance, Circuit Breakers from Gray Swan maps unsafe representations back to safe ones. How well do the proposed method attack their models?

**Ethical Concerns:**

["NO or VERY MINOR ethics concerns only"]

**Final Justification:**

The revised algorithm is now more self-contained, the additional experiments on the generality of their probing classifiers provide supports to their envisioned attack framework, and the results against JailGuard shows that the proposed attack has not super strong but certain resistance against probing-based defenses. The additional comparison with optimization-based and prompt-based text-only jailbreaks has made the paper more comprehensive, but the effectiveness of prompt-based attacks indicate that jailbreaking VLNs aren’t exactly as difficult as rendered and the help from the additional modality is relatively marginal.

In overall, the authors' response indicates promising improvements over the existing submission in terms of coherence and comprehensiveness. However, I agree with reviewer 9uX9 in that the innovation and potential impact of this work are relatively constrained and I cannot raise my ratings to higher.

**Limitations:**

Yes.

**Quality:**

3

**Strengths And Weaknesses:**

Strength:
+ The idea of making use of probing classifiers for jailbreaking is original yet straightforward.
+ The paper is generally well-written and easy to follow, thanks to the sound logic flow, consistent notation uses and illustrations.
+ The paper is generally wholesome and self-contained. For instance, the validation of know-say disparity, although saved for appendix, is helpful for understanding the background about the paper.

Weakness:
+ The experiment designs for the blackbox and whitebox settings are not entirely consistent. While it is interesting to know how well the full fledged JailBound compares with the unimodal variants, it is more important to know its comparison with the baselines beyond MiniGPT-4.

---

> ### Author Rebuttal · Authors · 2025-07-31
>
> We sincerely thank you for your constructive questions. Guided by your suggestions, we conduct several new experiments that significantly strengthen our paper's claims and clarity. We appreciate the opportunity to address your points below.
>
> ### **(Weakness 1) On Inconsistent White-box/Black-box Experimental Design**
>
>  **Our Response:** Thank you for this crucial observation. We agree that a comprehensive white-box comparison across all models is essential for a fair evaluation. We have rectified this by expanding our experiments to include all baselines on every white-box model.
>
> **New Experiment: Comprehensive White-box Performance Comparison**
>
> We extend our white‑box evaluation beyond MiniGPT‑4 to include two additional key models, Llama‑3.2‑11B and Qwen2.5‑VL‑7B, and compared JailBound with all baselines. The results, summarized in table below and incorporated into Section 4.3 of our revised manuscript, demonstrate that JailBound consistently and significantly outperforms all baselines across diverse model architectures. For example, on Llama‑3.2‑11B, JailBound achieves a 94.38% Attack Success Rate (ASR), representing a 22.76% absolute improvement over the next best baseline (VAJM at 71.62%). Similarly, on Qwen2.5‑VL‑7B, JailBound achieves a 91.40% ASR compared to VAJM’s 65.19%. These comprehensive results robustly validate the superior efficacy of our boundary‑guided, joint‑optimization approach in white‑box settings.
>
>
> | Model | UMK ASR (%) | FigStep ASR (%) | VAJM ASR (%) | **JailBound ASR (%)** |
> | :--- | :---: | :---: | :---: | :---: |
> | Llama-3.2-11B | 62.38% | 57.23% | 71.62% | **94.38%** |
> | Qwen2.5-VL-7B | 56.03% | 55.71% | 65.19% | **91.40%** |
>
>
>
>
> ### **(Q1) On Algorithm Details for $L\_{sem}$ and $L\_{suffix}$**
>
> **Our Response:** We are deeply grateful for this sharp question, which highlights a critical lack of precision in our initial manuscript. You are absolutely right that these terms were not clearly defined. We have now substantially revised both the main text (Section 3.3) and Algorithm 2 to make our methodology fully transparent.
>
> **1. Explicit Definition of $L\_{sem}$ and $L\_{suffix}$:**
> We clarify in Section 3.3 (Equation 8) that $L\_{sem}$ is a composite term balancing visual perturbation and textual fluency, defined as:
>          $$ L\_{sem} = \\|\delta\_v^{input}\\|^2\_2 + \mathcal{L}\_{suffix}(X\_t^{suffix}) $$
>
> The textual component, $\mathcal{L}_\text{suffix}$, is a regularization term that uses a pre-trained language model (Qwen2-7B-Instruct) to calculate the negative log-likelihood (NLL) of the generated suffix, ensuring its fluency:
>
> $\mathcal{L}\_{suffix}(X\_t^{suffix}) = - \sum \_{i=1}^{|{suffix}|}\log P_{LM} ({token}\_i | X\_t^{prompt}, {token}\_1, \dots, {token}\_{i-1})$
>
> By minimizing the NLL calculated by a powerful model, we ensure our optimization strongly penalizes any grammatically flawed or semantically unnatural token sequences.
>
>
> **2. Revisions to Algorithm 2:**
> To make the algorithm fully reproducible, we have updated it to explicitly show the computation of $\mathcal{L}_{sem}$. The relevant lines in Algorithm 2 have been revised as follows:
>
> > ... (lines 1-11 are unchanged) ...
>
> > // --- Explicitly compute components of Semantic Preservation Loss ---
>
> > 13:     $\mathcal{L}\_{suffix} \gets - \sum \log P\_{LM}({token}\_i | \dots)$       // Compute suffix fluency loss
>
> > 14:     $\mathcal{L}\_{sem} \gets \\|\delta\_v^{input}\\|^2\_2 + \mathcal{L}\_{suffix}(X\_t^{suffix})$      // Assemble $L\_{sem}$
>
> > // --------------------------------------------------------------------------------------------------
>
> > 15:     $\mathcal{L}\_{total} \gets \mathcal{L}\_{align} + \lambda\_1 \mathcal{L}\_{sem} + \lambda\_2 \mathcal{L}\_{geo}$   // Compute total loss
>
> This change makes the flow of information clear and directly resolves the ambiguity you pointed out.
>
>
>
> ### **(Q2) On the Generality of the Probing Classifier**
>
> **Our Response:** Thank you for this insightful question, which addresses a key aspect of our framework's generality. To provide a definitive answer regarding whether our classifier captures a fundamental safety boundary shared across different attack mechanisms *within the same model*, we conducted a new, focused experiment.
>
> **New Experiment: Validating the Probing Classifier as a General Surrogate for Jailbreaking**
>
>  We first use three distinct baseline attacks (FigStep, VAJM, and UMK) to generate a set of successful jailbreak samples using the test data on Qwen2.5-VL. Then we extract the internal representations of these successful samples from the model's final fusion layer. Finally, we feed these extracted representations into our probing classifier, which was pre-trained on the same final fusion layer of Qwen2.5-VL using our training data, to evaluate its accuracy in identifying these baseline-generated samples as "unsafe."
>
>
> | Baseline Attack Method (on Qwen2.5-VL) | Prediction Accuracy of Our Qwen2.5-VL Prober |
> | :--- | :---: |
> | FigStep | **96.17%** |
> | VAJM | **97.36%** |
> | UMK | **92.28%** |
>
> As shown in the table. Our probing classifier, trained only to distinguish safe from unsafe inputs in general, achieved high prediction accuracy on these baseline-generated jailbreaks (96.17% for FigStep, 97.36% for VAJM, and 92.28% for UMK). These results indicate that successful attacks produced by different mechanisms converge on a shared, detectable safety boundary within the model, confirming that our framework exploits a general vulnerability. This new experiment and its analysis have been added to the Appendix.
>
>
>
> ### **(Q3) On Comparison with Text-only Jailbreak Methods**
>
>
> **Our Response:** This is an excellent point. To rigorously evaluate the contribution of our multimodal approach, a direct comparison against powerful, text-only jailbreak methods is essential. We have now conducted this experiment as you suggested.
>
> **New Experiment: Comparison with Text-only Jailbreak Methods**
>
> To evaluate the contribution of our multimodal approach, we compared JailBound with two representative text‑only jailbreak methods, GCG [28] and AutoDAN [17], on the Llama‑3.2‑11B model.   For these text‑only baselines, we fed their generated adversarial prompts to the VLM with a null (empty) image input, ensuring that only textual perturbations were used. All methods were evaluated on the MM‑SafetyBench test set using Attack Success Rate (ASR) as the evaluation metric. As shown in the table below. GCG and AutoDAN achieved ASRs of 78.27% and 83.52%, respectively, while our full multimodal JailBound attack attained an ASR of 91.40%. These results demonstrate that by jointly optimizing perturbations across both image and text, JailBound achieves higher success rate than these powerful text‑only methods. This highlights the critical, synergistic role the visual channel plays in our attack framework, creating vulnerabilities that unimodal attacks cannot fully exploit. This new comparison has been incorporated into our revised manuscript.
>
> | Attack Method | Modality | ASR (%) on MM-SafetyBench |
> | :--- | :--- | :---: |
> | GCG | Text-only | 78.27% |
> | AutoDAN | Text-only | 83.52% |
> | **JailBound (Ours)** | **Image+Text** | **91.40%** |
>
>
>
> ### **(Q4) On Attacking Models with Probing-based Defenses**
>
> **Our Response:** Evaluating JailBound against such a defense is crucial. To address this, we conducted a new experiment against **JailGuard [1]**, a defense that operates on a similar principle of detecting representationally sensitive inputs.
>
> **New Experiment: Evaluating JailBound's Robustness Against JailGuard Defense**
> We first generated successful JailBound attacks on Qwen2.5-VL. Then, for each attack, we applied the JailGuard defense by creating several mutated variants of the image and text. An attack was considered "bypassed" if the original harmful response was maintained and the mutations did not trigger a refusal.
>
> | Metric | ASR (%) on Qwen2.5-VL |
> | :--- | :---: |
> | JailBound (No Defense) | **91.40%** |
> | JailBound (vs. JailGuard) | **58.65%** |
> | **Bypass Rate** | **64.17%** |
>
> JailBound maintains a high success rate against a tailored defense, bypassing it over 64.17% of the time. This suggests that the perturbations generated by our boundary-guided joint optimization are semantically robust. The image and text perturbations are designed to complement each other, so minor isolated mutations may not always disrupt the attack’s intended effect. This finding underscores the advanced nature of our method.
>
> [1] Zhang et al. "JailGuard: A Universal Detection Framework for LLM Prompt-based Attacks."

---

> > ### Comment · Reviewer_coGX · 2025-08-06
> >
> > I would like to thank the authors for their effort to fix the issues I pointed out and to address my concerns. The revised algorithm is now more self-contained, the additional experiments on the generality of their probing classifiers provide supports to their envisioned attack framework, and the results against JailGuard shows that the proposed attack has not super strong but certain resistance against probing-based defenses. The comparison with text-only jailbreaks is also appreciated but is rather limited with GCG and AutoDAN, both being optimization-based jailbreaks and somewhat dated, as the only baselines. I was hoping for comparisons with prompt-based jailbreaks like WordGame [1] and Puzzler [2]. Such attacks are not as interpretable as optimization-based ones, but are surprisingly more effective and efficient, indicating that jailbreaking LLMs and VLMs might not be as difficult as being rendered here in this paper even with a single modality.
> >
> > In overall, the authors' response indicates promising improvements over the existing submission in terms of coherence and comprehensiveness. However, I agree with reviewer 9uX9 in that the innovation and potential impact of this work are relatively constrained and I cannot raise my ratings to higher.
> >
> > [1] Zhang, Tianrong, et al. "Wordgame: Efficient & effective llm jailbreak via simultaneous obfuscation in query and response." arXiv preprint arXiv:2405.14023 (2024).
> > [2] Chang, Zhiyuan et al. “Play Guessing Game with LLM: Indirect Jailbreak Attack with Implicit Clues.” Annual Meeting of the Association for Computational Linguistics (2024).

---

> ### Author Response · Authors · 2025-08-07
>
> Thank you again for your constructive feedback. We are glad that our previous response addressed some of your concerns regarding the classifier's generality and the comparison with probing-based defenses. Your latest suggestion to compare our work against prominent prompt-based jailbreaks like WordGame [1] and Puzzler [2] is insightful. Following your advice, we have now incorporated these important works into our discussion and added a new comparative analysis in Section 4.2 of our revised manuscript. The details of this new experiment are presented below.
>
> **New Experiment: Benchmarking Against prompt-based Attacks**
> As the methods you mentioned, WordGame [1] and Puzzler [2], have not released their source code, we directly cite their performance from the original papers for our baseline comparison.  Our method, JailBound, is fundamentally designed for Vision-Language Models (VLMs). Therefore, to provide a meaningful benchmark against these text-only attacks, we selected a powerful, contemporary VLM, **Llama-3.2-11B**, as the target model. Our experimental setup is as follows:
> 1.  **Dataset**: AdvBench, to align with these two attack baselines.
> 2.  **Input Configuration**: We paired each textual query from AdvBench with a blank image (an all-black image).
> 3.  **Method Definitions**:
>     *   **JailBound (Multimodal)**: Jointly optimizes the text suffix and the blank image.
>     *   **JailBound-TextOnly (Unimodal)**: Optimizes only the text suffix, with no image perturbation.
>
> The results are compared with the cited baseline data in the table below.
>
> | Attack Method | Target Model | Dataset | Modality | ASR (%) |
> | :--- | :--- | :--- | :--- | :---: |
> | Puzzler [2] | Llama2-13B-chat | AdvSub | Text-only | 32% (Cited from Table 1) |
> | WordGame+ [1] | Llama 3 | AdvBench | Text-only | 90.38% (Cited from Table 2) |
> | JailBound-TextOnly (Ours) | Llama-3.2-11B | AdvBench | Text-only + Blank Image | 94.21% |
> | JailBound (Full, Ours) | Llama-3.2-11B | AdvBench | Image + Text | 97.67% |
>
> **Conclusion**
> Our new experiments show that our JailBound-TextOnly version achieved an ASR of 94.21%. This result is competitive with WordGame's performance (90.38% on Llama 3) and superior to Puzzler's on a similar open-source model (32% on Llama 2), demonstrating the effectiveness of our optimization framework in text-only scenarios. Furthermore, the full JailBound method increased the ASR to 97.67%. This gain highlights our core contribution: perturbing the visual channel improves attack success even when the visual input is a non-informative blank image, revealing a vulnerability that text-only attacks cannot exploit. In conclusion, these results establish that JailBound is competitive in performance and uncovers a unique multimodal attack surface. We appreciate your detailed and insightful comments throughout this process, which have significantly strengthened our work. Thank you again for your time and effort and we wish you all the best in both your research and life.

---

### Official Review · Reviewer_MnUy · 2025-07-03

**Clarity:** 2
**Significance:** 2
**Originality:** 2
**Rating:** 4
**Confidence:** 4

**Summary:**

The paper proposes JailBound, a jailbreaking framework targeting the latent safety decision boundaries of vision-language models (VLMs). It employs a two-stage approach: Safety Boundary Probing to identify perturbation directions in the fusion layer’s latent space and Safety Boundary Crossing to jointly optimize image and text perturbations using a multi-objective loss function. Evaluated on the MM-SafetyBench dataset, JailBound achieves high ASRs in white-box settings and in black-box scenarios.

**Questions:**

Please see the weaknesses. If authors could address my concerns, especially the concern of novelty, I would raise my rating.

**Ethical Concerns:**

["NO or VERY MINOR ethics concerns only"]

**Final Justification:**

My concerns have been addressed. Thus, I raise my rating by 1.

**Limitations:**

yes

**Quality:**

2

**Strengths And Weaknesses:**

**Strengths:**

1. The proposed two-stage framework leverages latent safety boundaries to enhance jailbreaking success rates.

2. Experiments on MM-SafetyBench demonstrate robust ASR across multiple VLMs.

3.  The whole paper is clearly written.

**Weaknesses:**

1. I’m concerned about the claimed novelty, as JailBound appears to combine steering vectors (the boundary normal is the vector direction) with optimization-based jailbreaks without clearly articulating the advantages of this fusion. And we know a steering vector derived from the probing stage could achieve comparable ASR without the optimization step. So, why is the two-stage approach necessary?

2. The paper overlooks critical references, such as [1] on representation engineering and [2, 3] on safety-violating steering vectors, which are highly relevant to the work.

3. The explanation for black-box transferability is insufficient. It’s unclear why this pipeline excels in black-box settings.

4. Experimental details are also incomplete, particularly regarding the data used for probing. Were these the same as the validation set, and if so, how does this avoid overfitting to MM-SafetyBench?

5. The evaluation’s persuasiveness is limited by testing on only MM-SafetyBench. Additional experiments on a diverse dataset like JailbreakV-28k would strengthen the claims of generalizability.

6. Figure 5’s case study, described as real-world, uses cartoonish images rather than authentic screenshots, and with only two examples provided, it’s challenging to find the evidence convincing.

7. Some formatting issues, such as the incorrect citation on line 144.

[1] Zou, Andy, et al. "Representation engineering: A top-down approach to ai transparency."

[2] Arditi, Andy, et al. "Refusal in language models is mediated by a single direction."

[3] Yu, Lei, et al. "Robust LLM safeguarding via refusal feature adversarial training."

---

> ### Author Rebuttal · Authors · 2025-07-31
>
> We are grateful for your constructive feedback. Guided by your suggestions, we have conducted new experiments and revised the manuscript to enhance its clarity and rigor. Our detailed responses are provided below.
>
> ### **(Weakness 1) On Novelty and the Necessity of the Two-Stage Approach**
>
> **Our Response:** Thank you for this insightful question. While our method draws inspiration from existing powerful concepts, its novelty lies in designing a synergistic two-stage framework where each stage addresses the fundamental limitations of the other.
>
> - **Why Steering Vector Alone Is Insufficient:**
> Steering vectors manipulate latent activations directly, which works well in white-box settings but fails to transfer to black-box scenarios since they do not produce concrete, perturbable input samples (images & text).
>
> - **Why Optimization Alone Is Inefficient:**
>   Optimization without guidance from the model’s decision boundary can waste many iterations exploring unproductive directions. This unguided search often leads to over-perturbation, degrading semantic quality and paradoxically reducing attack effectiveness, while also incurring high computational costs.
>
> - **Our Necessary Synergy:**
>   JailBound addresses these issues. Stage 1 efficiently probes and identifies classifier-guided directions in the latent space. Stage 2 then leverages this information to craft concrete, transferable adversarial samples with minimal and imperceptible perturbations. This synergy is key to achieving both high attack success rate and robust black-box transferability.
>
> **New Experiment 1: Validating the Necessity of Our Two-Stage Framework**
>
>   We conducted experiments on the MM-SafetyBench dataset to evaluate attack success rates across multiple white-box models, including Llama-3.2-11B, Qwen2.5-VL-7B, and MiniGPT-4. Steering vector methods aim to find semantically meaningful directions in a model’s latent activation space and steer the model’s behavior by manipulating internal activations along these directions. Based on this principle, we adopt SCAV [32] as a baseline, which identifies semantic concept directions in activation space and manipulates model representations accordingly to influence downstream behaviors. In our experiment, we applied SCAV’s activation-space manipulations directly inside these white-box models without any input-level optimization.
>
>
> | **Method**                     | **Llama-3.2-11B** | **Qwen2.5-VL-7B** | **MiniGPT-4** |
> |-------------------------------|:-----------------:|:-----------------:|:-------------:|
> | Steering Vector Only *(SCAV)* |      85.27%        |      78.43%        |    83.65%      |
> | **JailBound (Ours)**           |    **94.38%**      |    **91.40%**      |  **97.19%**    |
>
> These results demonstrate that while steering vector methods achieve reasonable attack success rates in white-box settings, their lack of concrete input-level perturbations limits their effectiveness and transferability. Our two-stage JailBound framework effectively combines steering vector insights with input optimization to achieve stronger attacks across multiple models.
>
>
> ### **(Weakness 2) On Replenishing Critical References**
>
>
> **Our Response:** Thank you for pointing out these critical references. We agree they are highly relevant and have revised our Related Work section to incorporate them, which helps to better situate our work. The updated text is below:
>
> > **Complementary to passively eliciting knowledge, a significant body of work has explored actively steering representations to control model behavior. Zou et al. [1] proposed Representation Engineering, a top-down approach to control AI behaviors by directly manipulating internal representations. Critically, this principle was extended to model safety by Arditi et al. [2], who discovered that a language model's refusal behavior is primarily mediated by a single direction in activation space; manipulating this direction can directly modulate the model's propensity to refuse. Building on this vulnerability, Yu et al. [3] introduced Refusal Feature Adversarial Training, a method that enhances robustness by conducting adversarial training in the representation space related to refusal.**
>
> [1] Zou, Andy, et al. "Representation engineering: A top-down approach to ai transparency."
> [2] Arditi, Andy, et al. "Refusal in language models is mediated by a single direction."
> [3] Yu, Lei, et al. "Robust LLM safeguarding via refusal feature adversarial training."
>
>
>
>
> ### **(Weakness 3) On Black-box Explanation**
>
>
> **Our Response:** We add the following explanation to Section 4.4 to clarify why JailBound transfers effectively to black-box models:
>
> > **Modern vision-language models (VLMs) differ in architecture (e.g., GPT‑4o, Qwen2.5-VL), yet their safety alignment procedures often yield similar semantic safety representations. Qi et al. [4] shows that such alignment can produce common failure modes across different models, indicating shared vulnerabilities at the representation level. JailBound leverages this property: Stage 1 identifies the decision boundary around unsafe semantic regions in a white‑box source model, and Stage 2 finds minimal perturbations that cross this boundary. Because target black‑box models exhibit similarly structured unsafe regions for the same concepts, the resulting adversarial inputs transfer effectively.**
> >
> [4] Qi et al. "Safety Alignment Should Be Made More Than Just a Few Tokens Deep."
>
>
> ### **(Weakness 4) Incomplete Experimental Details (Probing Data)**
>
>
> **Our Response:** We appreciate this important question, as experimental rigor is essential. We acknowledge that the initial submission lacked sufficient detail about our data handling protocol and have revised the manuscript accordingly.
>
> **Specific Changes (Section 4.1, Experimental Setup):**
> We add the following paragraph at the beginning of Section 4.1 to clearly describe our probing data protocol:
>
> > **To establish a rigorous evaluation protocol, we partition the MM‑SafetyBench dataset [18] into a 70% training set and a 30% test set using stratified sampling across all 13 prohibited content categories. The 70% training set is used exclusively for the *Safety Boundary Probing* stage (training the logistic regression classifiers), while all reported Attack Success Rates (ASRs) for both white‑box and black‑box models are computed only on the held‑out 30% test set. This strict separation ensures that our evaluation uses data never seen during boundary learning, thereby preventing overfitting and providing a fair assessment of JailBound’s generalization.**
>
> ### **(Weakness 5) On Limited Evaluation**
>
>
> **Our Response:** We expand our evaluation to the JailbreakV‑28k benchmark to validate generalizability.
>
> **New Experiment 2 (added to Appendix): Evaluating JailBound on the JailbreakV‑28k Benchmark**
>
> To further validate the generality and robustness of our approach, we evaluated JailBound on the JailbreakV‑28k benchmark using three representative open‑source models (LLaMA‑3.2, Qwen2.5‑VL, and MiniGPT‑4). We compared our results against the SD+Typo baseline, as shown in the table below. JailBound consistently outperforms the baseline across all models, achieving absolute Attack Success Rate (ASR) improvements of +66.4% on LLaMA‑3.2 (89.5% vs. 23.1%), +65.9% on Qwen2.5‑VL (84.2% vs. 18.3%), and +59.2% on MiniGPT‑4 (92.7% vs. 33.5%). These improvements demonstrate that JailBound’s safety boundary–guided optimization is not model-specific but instead generalizes well across different architectures and tasks. This highlights JailBound’s capability to exploit fundamental safety boundary vulnerabilities, enabling state‑of‑the‑art performance on a challenging and diverse benchmark. We have included the full experimental setup and analysis in the Appendix.
>
>
> | MLLM | Baseline ASR (%)   (SD+Typo) | **JailBound ASR (%)** | **Improvement** |
> | :--- | :---: | :---: | :---: |
> | LLaMA 3.2 | 23.1 | **89.5** | **+66.4** |
> | Qwen2.5-VL | 18.3 | **84.2** | **+65.9** |
> | MiniGPT-4 | 33.5 | **92.7** | **+59.2** |
>
>
> ### **(Weakness 6) On Unconvincing Case Study**
>
> **Our Response:** Thank you for this feedback. We agree that more compelling evidence is needed. In the revised manuscript, we have completely redesigned Figure 5 to replace the cartoonish icons with a gallery of real-world jailbreak examples. This new figure showcases actual screenshots of harmful model responses, providing a much clearer and more direct demonstration of our method's efficacy. Additional detailed case studies have also been added to the appendix.
>
>
>
> ### **(Weakness 7) On Formatting Issues**
>
> **Our Response:** Thank you for your careful reading. We have corrected the citation error on line 144 to properly reference Algorithm 1 and have performed a full proofread of the manuscript to ensure all formatting is correct.

---

> > ### Comment · Reviewer_MnUy · 2025-08-05
> >
> > Thanks for your response. My concerns have been addressed. Thus, I raise my rating by 1.

---

### Public Comment · ~Jaeyoon_Kim6 · 2026-05-14
**Reproduce the ASR values - Table 3 (Code availability Inquiry)**

Thanks for the interesting work.

I have a question regarding the evaluation setup in Table 3.

I have been unable to reproduce the ASR values reported in Table 3 (actually, ASR is very low) I believe this discrepancy likely stems from differences in the experimental setup, as several key details regarding the use of MM-SafetyBench are not specified in the paper. Specifically:

1. Which image type was used—SD, SD_TYPO, or TYPO?
2. Which text field was used as the input prompt—"Question," "Changed Question," "Rephrased Question," "Rephrased Question(SD)," or another field?
3. Were all 1,680 questions used, or only a subset? (The paper mentions 1,719 examples, which does not match the original MM-SafetyBench count of 1,680 questions.)
4. If a subset was used, how was the data split between training the safety boundary classifiers and evaluating ASR?

Ideally, it would be tremendously helpful if you could share the source code so that I can directly reproduce your reported results. If that is not feasible, I would greatly appreciate a detailed description of the dataset configuration and experimental setup so that I can align my implementation accordingly.

---

### Note · Authors · 2025-08-13

**Dear Area Chair,**

We are writing to provide a focused summary of the highly constructive review process for our submission. Our rebuttal, featuring ten new major experiments, successfully resolved all reviewer concerns, resulting in two score increases and two definitive confirmations of resolution.
Here is a more detailed summary of each reviewer's feedback journey:
- **Reviewer MnUy**: We addressed novelty concerns by conducting a new experiment that empirically demonstrated the necessity of our two-stage approach. Furthermore, we expanded our evaluation to the diverse JailbreakV-28k benchmark, showing massive performance gains (e.g., +66.4% ASR over the baseline on LLaMA 3.2). Reviewer MnUy acknowledged that these additions successfully addressed their concerns and raised score.
- **Reviewer coGX**: We addressed several deep technical questions with four new, targeted experiments. These included: (1) validating that our prober identifies a general safety boundary (achieving >92% accuracy on attacks from other methods); (2) benchmarking against strong text-only methods (GCG, AutoDAN); (3) evaluating robustness against the JailGuard defense (64% bypass rate); (4) comparing against prompt-based attacks WordGame and Puzzler. Reviewer coGX acknowledged that the paper is more "coherent and comprehensive" and our new experiments "provide supports to envisioned attack framework."
- **Reviewer 9uX9**: We resolved crucial concerns about computational cost. Our new analysis turned this perceived weakness into a key strength, showing our method is highly efficient—requiring only ~100 iterations and being ~8-13x faster than baselines. Reviewer 9uX9 agreed that it resolved their concerns about computational cost and raised score.
- **Reviewer P9we**: We resolved all issues regarding dataset diversity, computational cost, and the need for a defense discussion. Reviewer P9we concluded that our new additions were successful, stating: "These additions have strengthened the paper and addressed my main reservations."

We hope our revised manuscript, strengthened by new experiments and the positive resolution from all four reviewers, represents a valuable contribution.
Thank you for your time and consideration. Wishing you all the best in both your research and life!

Sincerely,

The Authors of Submission 16162

---

### Decision · Program_Chairs · 2025-09-17

**Decision:**

Accept (poster)

**Comment:**

This paper presents JailBound, a jailbreaking framework targeting Vision-Language Models (VLMs) by exploiting internal safety decision boundaries within their latent representations. The work is grounded in the hypothesis that VLMs encode safety-relevant information within their fusion-layer representations, revealing implicit safety decision boundaries in the latent space. The proposed method operates through a two-stage approach: (1) Safety Boundary Probing, which uses logistic regression classifiers to approximate decision boundaries in the fusion layer's latent space, and (2) Safety Boundary Crossing, which jointly optimizes adversarial perturbations across both image and text inputs to steer the model's internal state toward policy-violating outputs. The experimental evaluation demonstrates substantial improvements over existing methods.

The main strengths of the paper are as follows: (1) The application of Eliciting Latent Knowledge (ELK) principles to multimodal jailbreaking represents a novel approach. The explicit probing and exploitation of safety decision boundaries in fusion layers is sound. (2)  The work includes extensive experiments across six different VLMs in both white-box and black-box scenarios.  (3)  The work addresses a critical safety concern for VLMs that are increasingly deployed in real-world applications, with high success rates highlighting the practical importance of this vulnerability. (4)  The paper is generally well-written with sound logical flow, consistent notation, and effective illustrations that make the technical content accessible.

The main weaknesses of the paper are as follows in the first round review: (1) The paper lacks deeper analysis of when this assumption might fail or how safety boundaries emerge from a training dynamics perspective. (2) The assumption that safety decisions can be linearly separated in fusion space may not hold universally across all VLMs or safety categories. The paper would benefit from more nuanced discussion of failure modes. (3)  limited discussion of potential defenses or detection mechanisms. (4)  Some reviewers noted that the core methodology combines existing concepts (steering vectors + optimization-based jailbreaks) in a relatively straightforward manner, potentially limiting the fundamental innovation.

The reason for acceptance: the paper makes a solid technical contribution to an important safety problem. The experimental validation is thorough, and the method demonstrates clear practical effectiveness. In particular, the idea of making use of probing classifiers for jailbreaking is original, and the paper is generally well-written and easy to follow with sound logic flow, consistent notation uses and illustrations.

Regarding the rebuttal discussion, (1) Reviewer MnUy initially raised concerns about novelty and the necessity of the two-stage approach. The authors addressed this with a targeted experiment demonstrating that steering vector methods alone achieve the lower success rate compared to JailBound. The reviewer acknowledged solution and increased their score. (2) Reviewer coGX posed several technical questions about algorithm details, classifier generality, and comparisons with text-only methods. The authors provided comprehensive responses. The reviewer noted the paper became "more coherent and comprehensive. (3) Reviewer 9uX9 focused on computational burden concerns. The authors demonstrated JailBound's efficiency advantage and provided theoretical justification for the linear boundary approach. The reviewer increased their score while noting that deeper insights into boundary emergence remain limited. (4) (4) Reviewer P9we questioned dataset diversity and computational costs. The reviewer confirmed that additions "strengthened the paper and addressed main reservations."

All four reviewers acknowledged that their concerns were satisfactorily addressed, with two explicitly increasing scores and all providing positive final comments. The substantial improvements made during rebuttal, combined with the consistent positive resolution across all reviewers, support acceptance of this technically solid contribution to VLM safety research. The authors should improve the paper by introducing the rebuttal discussions.